# Patterns, Predictors, and Prognostic Value of Skeletal Muscle Mass Loss in Patients with Locally Advanced Head and Neck Cancer Undergoing Cisplatin-Based Chemoradiotherapy

**DOI:** 10.3390/jcm10081762

**Published:** 2021-04-18

**Authors:** Najiba Chargi, Inge Wegner, Navid Markazi, Ernst Smid, Pim de Jong, Lot Devriese, Remco de Bree

**Affiliations:** 1Department of Head and Neck Surgical Oncology, Division of Imaging and Oncology, University Medical Center Utrecht, Heidelberglaan 100, 3584 CX Utrecht, The Netherlands; N.Chargi-2@umcutrecht.nl (N.C.); h.a.markazi@students.uu.nl (N.M.); 2Department of Otorhinolaryngology and Head and Neck Surgery, University Medical Center Groningen, 9713 GZ Groningen, The Netherlands; wegner.inge@gmail.com; 3Department of Radiation Oncology and Nuclear Medicine, Division of Imaging and Oncology, University Medical Center Utrecht, Heidelberglaan 100, 3584 CX Utrecht, The Netherlands; E.J.Smid-2@umcutrecht.nl; 4Department of Radiology, Division of Imaging and Oncology, University Medical Center Utrecht and Utrecht University, Heidelberglaan 100, 3584 CX Utrecht, The Netherlands; P.deJong-8@umcutrecht.nl; 5Department of Medical Oncology, Division of Imaging and Oncology, University Medical Center Utrecht and Utrecht University, Heidelberglaan 100, 3584 CX Utrecht, The Netherlands; L.A.Devriese@umcutrecht.nl

**Keywords:** sarcopenia, head and neck cancer, body composition, skeletal muscle mass, muscle wasting, chemoradiotherapy, image-based analysis

## Abstract

Low skeletal muscle mass (SMM) is associated with toxicities and decreased survival in head and neck cancer (HNC). Chemoradiotherapy (CRT) may exaggerate loss of SMM. We investigated the changes in SMM, their predictors, and prognostic impact of SMM in patients treated with CRT between 2012 and 2018. Skeletal muscle area (SMA) segmentation was performed on pre- and post-CRT imaging. Observed changes in SMM were categorized into: (I) Stable, (II) moderate gain (III), moderate loss, (IV) large gain, and (V) large loss. In total, 235 HNC patients were included, of which 39% had stable SMM, 55% moderate loss, 13% moderate gain, 0.4% large loss, and 0.4% large gain of SMM. After CRT, SMA decreased compared to pre-CRT (31.6 cm^2^ versus 33.3 cm^2^, *p* < 0.01). The key predictor was a body mass index (BMI) of ≥30 kg/m^2^ (OR 3.6, 95% CI 1.4–9.3, *p* < 0.01). Low SMM at diagnosis (HR 2.1; 95% CI 1.1–4.1, *p* = 0.03) and an HPV-positive oropharyngeal tumor (HR 0.1; 95% CI 0.01–0.9, *p* = 0.04) were prognostic for overall survival. Changes in SMM were not prognostic for survival. Loss of SMM is highly prevalent after CRT and a high BMI before treatment may aid in identifying patients at risk.

## 1. Introduction

Head and neck cancer (HNC) accounts worldwide for approximately 550,000 cases annually [1]. Locally advanced HNC (LA-HNC) is the most prevailing clinical manifestation of HNC and has poor prognosis with a 5-year disease-free survival (DFS) of approximately 40–50%. Although the addition of the cytotoxic compound cisplatin to radiotherapy (RT) improved 5-year PFS from 36% to 47% and 5-year overall survival (OS) from 40% to 53%, it also caused a significant increase in severe functional mucosal adverse effects from 21% to 40% [2]. Due to the increased risk of side-effects, full compliance of chemoradiotherapy (CRT) is seen in only about two-thirds of the initially eligible patients [3,4,5]. Ongoing research evaluates and compares different systemic treatment regimens and novel therapeutic approaches with consideration of potential patient-related (i.e., HPV status) and treatment-related factors (i.e., dose regimen) in order to improve treatment tolerance and survival in LA-HNC patients.

An emerging patient-related predictive and prognostic factor in the management of HNC is patients’ skeletal muscle mass (SMM). SMM quantification can be easily performed with the use of computed tomography (CT) or magnetic resonance imaging (MRI) images, which are routinely performed in the diagnostic work-up prior to treatment. Low SMM is common in HNC and especially in LA-HNC patients [6]. Patients with LA-HNC frequently experience dysphagia due to tumor site and adverse effects caused by CRT. This leads to weight loss and nutritional deficiencies which are the major contributors to low SMM.

Low SMM has been shown to be a significant predictive factor for failure of the treatment plan due to toxicities in various types of cancer [7,8,9,10]. In addition, in LA-HNC patients, low SMM at diagnosis has been shown to be predictive for platinum dose-limiting toxicities [11,12]. Moreover, previous studies suggest that chemotherapy itself may induce SMM loss, also referred to as muscle wasting, in patients with cancer by increasing lipolysis and fatty acid B-oxidation [13]. It has also been suggested that patients with low SMM have higher blood levels of cytotoxic agents compared to patients without low SMM, which together with the previous mentioned mechanism may cause a vicious circle of muscle wasting [12]. In addition, various studies in other types of cancers have shown that loss of SMM during systemic chemotherapy is prognostic for decreased survival in patients with several types of cancer including colorectal and pancreatic cancer [14,15,16,17].

For HNC, several studies have shown that low SMM at diagnosis is prognostic for decreased survival [18,19,20,21]. However, little is known about the patterns and prognostic impact of changes in SMM after cisplatin-based CRT in LA-HNC patients. In HNC, one previous study investigated the prognostic impact of changes in SMM after (C)RT in patients with nasopharyngeal carcinoma and showed that loss of SMM was associated with decreased OS [22]. Nasopharyngeal carcinoma is, however, a distinctive entity in HNC. Furthermore, patients were treated with different treatment strategies (induction CRT as well as concurrent CRT) and SMM segmentation was performed on CT scans with a wide time interval (median: 110 days, range 41–1083 days). Regarding squamous cell carcinomas of other anatomical head and neck subsites, no evidence has been published yet. If loss of SMM after CRT is indeed a prognostic factor, it could be used as an objective measurement tool for decision making and it may offer the opportunity for timely therapeutic intervention to potentially reverse muscle wasting.

Therefore, this study evaluated the patterns and predictors of changes in SMM in LA-HNC patients treated with cisplatin-based CRT. In addition, the aim of this study was to determine if low SMM before CRT or loss of SMM after CRT have a prognostic impact on OS and DFS in LA-HNC patients.

## 2. Material and Methods

### 2.1. Ethical Approval

The design of this study was approved by the Medical Ethical Research Committee of the University Medical Center Utrecht, METC ID: 17–365/C, date of approval: January 2019. The requirement for informed consent from patients was waived because of its retrospective design.

### 2.2. Study Design

A retrospective cohort study was conducted. All patients diagnosed with LA-HNC and treated with cisplatin-based CRT in primary or postoperative setting between 2012 and 2018 in our tertiary referral center were screened for inclusion. Inclusion criteria for this study required that patients had CT or MRI imaging of the head and neck area within 1 month before CRT and follow-up CT or MRI imaging within 1 year after completion of CRT. Relevant demographic and clinical variables were retrieved from patients’ electronic medical records.

### 2.3. Therapy

Chemotherapy regimen consisted of three cycles of intravenous cisplatin-based chemotherapy on days 1, 22, and 43 of CRT. Chemotherapy dose was 100 mg/m^2^. CRT was given in a primary setting for patients with (technical or functional) irresectable LA-HNC and in a postoperative setting in case of positive resection margins and/or in the presence of extranodal tumor extension in resected lymph node metastases. Radiotherapy was administered in 35 fractions of 2 Gy to make a total dose of 70 Gy (primary setting), and in 33 fractions of 2 Gy to make a total dose of 66 Gy (postoperative setting).

### 2.4. Skeletal Muscle Measurements

Skeletal muscle area (SMA) was segmented using the Slice-O-matic software. Patients’ SMA was segmented on pre-CRT imaging and post-CRT imaging. At the level of the third cervical vertebra (C3), a single slice was used for SMA segmentation. The first slide to completely show the entire vertebral arc when scrolling through the C3 vertebra in caudal to cephalic direction was selected. For CT imaging, muscle area was defined as the pixel area between the radiodensity range of −29 and +150 Hounsfield units (HU), which is specific for muscle tissue [23]. For MRI, muscle area was manually segmented, and fatty tissue was manually excluded. The overall intraclass correlation coefficient (ICC) for the muscle SMA obtained by CT and MRI has shown to be excellent (ICC 0.9, *p* < 0.01) [24], and can therefore be used interchangeably for measuring CSA at the level of C3. The SMA was calculated as the sum of the delineated areas of the paravertebral muscles and both sternocleidomastoid muscles. When measurement of SMA of one musculus sternocleidomastoideus was not possible, due to, e.g., lymph node invasion or radical neck dissection, we calculated the SMA of the other sternocleidomastoid muscle and multiplied this by two. SMA at the level of C3 was first converted to SMA at the level of L3 using a previously published formula [25]. The SMA at the level of L3 was corrected for patients’ squared height to obtain the lumbar skeletal muscle mass index (LSMI).

The LSMI cutoff value for the diagnosis of low SMM chosen in this study was determined by calculating the log likelihood using a statistical method previously described by Williams et al. [26]. The cutoff value best associated with survival (lowest log-likelihood) was a LSMI ≤ 46.6 cm^2^/m^2^.

### 2.5. Skeletal Muscle Mass Changes

Relative changes of SMM were calculated by using the following formula:
Relative change of SMM = (SMA after CRT–SMA before CRT) / (SMA before CRT) × 100%.

Hereafter, the standard deviation (SD) of the relative changes in SMM was calculated as previously described by Brown et al. [27] in order to derive five categories of changes in SMM in the fifth to ninety-fifth percentiles:Stable changes in SMM: No change ±1 SD from baseline;Moderate gain in SMM: ≥1 SD to <2 SD of gain from baseline;Moderate loss in SMM: ≥1 SD to <2 SD of loss from baseline;Large gain in SMM: ≥2 SD from baseline;Large loss in SMM: ≥2 SD from baseline.

### 2.6. Survival

OS was defined as the time between the date of histologic diagnosis of LA-HNC and death, or date of last follow-up. DFS was defined as the time between the date of histologic diagnosis of LA-HNC and the date of pathologic confirmed recurrence or date of last follow-up, whichever occurred first.

### 2.7. Statistical Analysis

Data analysis was performed using IBM SPSS statistics 25. Demographic and clinical data were reported for the included patients. Baseline measures for these groups were described using descriptive statistics. Normally distributed variables were shown as means ± standard deviation (SD); non-normally distributed variables were shown as medians with an interquartile range (IQR). Normality was investigated using the Kolmogorov–Smirnov test. Independent sample Student’s *t*-tests were used to compare the means of normally distributed continuous variables with regard to presence or absence of low SMM. Categorical variables were described as frequencies with corresponding percentages. The Fisher exact test and Chi-square test were used, when appropriate, for analyzing differences between the frequencies of each categorical variable with regard to the presence or absence of low SMM and of muscle changes.

A logistic regression model was used for univariate and multivariate analysis of the predictors for loss of SMM (including patients from the groups of moderate and large loss of SMM) and a cox-proportional hazard model was used for the prognostic impact of low SMM at baseline and a loss of SMM (including patients from the groups of moderate and large loss of SMM) after CRT on OS and DFS. The time interval chosen in the Cox proportional hazard model was the time between pre-CRT and follow-up imaging for estimating the predictors for loss of SMM and the time interval between diagnosis and the date of the event (death, recurrence) for estimating the prognostic impact of SMM on OS and DFS. Covariates used in the multivariate analysis were selected based on clinical relevance and if the *p*-value in the univariate analysis was <0.05 or near significance <0.1. Clinical relevance was determined based on literature and expert opinion. Furthermore, the relationship between survival and SMM and change in SMM was visualized using Kaplan–Meier survival curves, including log-rank tests.

Statistical significance was evaluated at the 0.05 level using 2-tailed tests.

## 3. Results

### 3.1. Patients’ Characteristics

In total, 235 LA-HNC patients were identified who received cisplatin-based CRT between 2012 and 2018 and had evaluable pre-CRT and follow-up imaging of the head and neck area within 1 year. The median time between follow-up imaging and pre-CRT imaging was 6 months (IQR 5–9). The follow-up period of the included patients ranged from November 2012 until May 2019.

The clinical and demographic characteristics of the study population prior to initiation of CRT are presented in Table 1. The majority of patients was male (70%). Mean age at diagnosis was 59 years (SD 8) and the mean body mass index (BMI) was 24.5 kg/m^2^. Nearly half of the patients (49%) had mild comorbidities as evaluated by the Adult Comorbidity Evaluation 27 (ACE-27) score. Most patients were current/former smokers (82%) and/or consumed alcohol (83%), and 75% of patients had combined tobacco and alcohol use. Most patients had an Eastern Cooperative Oncology Group (ECOG) performance status of 1 (47%). The mean serum albumin levels of the included patients at diagnosis were 39.8 g/L (SD 4.6). Most patients had either a tumor located in the oral cavity (35%) or oropharynx (31%) of which the majority (60%) was not associated with human papillomavirus. A majority of the patients (83%) was diagnosed with a tumor, node, metastasis (TNM) stage IV tumor, and underwent CRT in a primary setting (71%).

At pre-CRT imaging, 141 patients (60%) had low SMM. As is shown in Table 1, patients with low SMM were significantly more likely to be female, to be of older age, to be (current/former) smokers, to have combined tobacco and alcohol use, and to have a BMI < 18.5 kg/m^2^.

At follow-up imaging, 149 patients (63%) were diagnosed with low SMM. Mean SMA at follow-up imaging (31.62 cm^2^, SD 8.69) was significantly lower than mean SMA at pre-CRT imaging (33.34 cm^2^, SD 9.11) (*p* < 0.01). Table 2 shows the SMM changes that occurred in the study population. Only 91 patients (39%) had stable SMM compared to pre-CRT. A rather large proportion of the study population (*n* = 129, 55%) had moderate loss in SMM compared to pre-CRT SMM and only 13 (6%) patients showed moderate gain in SMM compared to pre-CRT SMM. A minority of patients experienced large loss in SMM (*n* = 1, 0.4%) or large gain in SMM (*n* = 1, 0.4%).

Table 3 shows the characteristics of patients with stable SMM versus patients with loss in SMM (moderate and large loss) and gain in SMM (moderate and large gain). Patients with loss in SMM were more likely to have a BMI ≥ 25 kg/m^2^ compared to patients with stable SMM (*p* < 0.01).

### 3.2. Predictors of Loss in Skeletal Muscle Mass

Table 4 shows the logistic regression analysis of the predictors for loss of SMM. In univariate regression analysis, significant predictors for loss of SMM were a BMI ≥ 30 kg/m^2^ (odds ratio (OR) 2.9, 95% confidence interval (CI) 1.2–6.9), and an unknown primary tumor (OR 0.1, 95% CI 0.01–0.9). In multivariate logistic regression analysis, a BMI ≥ 30 kg/m^2^ (HR 3.6, 95% CI 1.4–9.3) remained a significant predictor for loss of SMM after CRT.

### 3.3. Survival: Overall Survival and Disease-Free Survival

During the follow-up period from November 2011 until May 2019, 86 (37%) patients died, and 72 (31%) patients developed a recurrence. The median OS was 22 months (IQR 12–39) and the median DFS was 19 months (IQR 9–35). Of the patients that died, 43 (50%) patients experienced a loss in SMM, 36 (42%) patients had stable SMM changes, and 7 (8%) patients gained SMM after treatment. Of the patients that developed a recurrence during follow-up, 36 (50%) patients had a loss in SMM, 31 (43%) patients had stable SMM changes, and 5 (7%) patients gained SMM. Although half of the patients who died or had a recurrence during follow-up experienced a loss in SMM, in univariate Cox regression analysis, no prognostic value of loss in SMM for OS nor DFS were found. Using stable or gain in SMM changes as the reference group, HRs for SMM loss were 0.9 (95% CI 0.6–1.4) and 0.8 (95% CI 0.5–1.3), respectively. Figure 1 shows the Kaplan–Meier OS and DFS curves for patients with loss in SMM versus no loss in SMM (stable SMM and muscle gain (moderate and large)).

As shown in Table 5, in univariate Cox regression analysis, low SMM prior to initiation of CRT (HR 2.1; 95% CI 1.1–4.1, *p* = 0.03) and an HPV-positive oropharyngeal tumor (HR 0.1; 95% CI 0.01–0.9, *p* = 0.04) were prognostic for OS. No significant prognostic impact of low SMM for DFS was seen (HR 1.4; 95% CI 0.9–2.3). Low SMM after treatment showed no prognostic value for OS (HR 1.4; 95% CI 0.9–2.2) nor DFS (HR 1.3; 95% CI 0.8–2.2). Figure 2 shows the Kaplan–Meier OS and DFS curves for patients with low SMM before CRT.

## 4. Discussion

This study is the first to evaluate the patterns of changes in SMM in LA-HNC patients treated with cisplatin-based CRT. After CRT, the majority of the patients (*n* = 129, 55%) had moderate loss of SMM and one patient (0.4%) had large loss of SMM after CRT. A minority of the patients (*n* = 13, 6%) had moderate gain of SMM after CRT, and one patient (0.4%) had large gain of SMM after CRT. Of the 235 LA-HNC patients who underwent CRT, only 91 patients (39%) had stable SMM. The mean SMA at follow-up (31.62 cm^2^, SD 8.69) was significantly lower than mean SMA at the initiation of CRT (33.34 cm^2^, SD 9.11). Although this finding was statistically significant, further research should be done to investigate its clinical significance and its relationship with BMI changes. The prevalence of low SMM before treatment was 78.7%.

Previous research has shown that low SMM is a significant negative prognostic factor for survival [19,21]. As reported in these studies, this study showed that low SMM prior to initiation of CRT was a negative prognostic factor for overall survival in univariate analysis, but this finding was not statistically significant when the HPV status was included in multi-variate analysis. An explanation for why we did not find a prognostic impact for overall survival in multivariate analysis might be that tumor stage and HPV status as prognostic factors outweigh the prognostic impact of low SMM. In a previous study in elderly HNC patients, we also showed that low SMM had prognostic impact in patients with stage I–III HNC, but lost its prognostic impact in patients with stage IV HNC [21]. In this study, we included patients with LA-HNC, stages III–IV, and it is possible that low SMM does not have a prognostic impact in this group of patients. The mechanism underlying the relation between low SMM and decreased survival is yet to be elucidated. Low SMM may impact survival by causing treatment-related toxicities, which may lead to ineffective cancer treatment. In a previous study, although low SMM was not prognostic for the whole group of HNC patients treated with primary CRT, patients who experienced cisplatin dose-limiting toxicity, significantly more frequently observed in patients with low SMM, had a worse prognosis [12]. Moreover, malignancies also cause a state of hyper catabolism and inflammation which negatively impacts SMM, causing a negative vicious circle [28].

The loss of SMM after treatment, also referred to as muscle wasting, is the net result of a combination of an imbalance between protein synthesis and protein degradation, cell death of muscle cells, and a decrease in the muscle’s capability of regenerating new muscle cells. Previous research also underlines the role of oxidative stress and inflammation in the development of muscle wasting [29].

In this study, being obese at diagnosis showed to be a significant predictive factor for loss of SMM in our study population. Although this may feel counterintuitive, BMI may mask an underlying unfavorable body composition, i.e., a patient may be overweight by a surplus of fatty tissue and still have low SMM. This combination of low SMM and a surplus of fatty tissue is also referred to in the literature as sarcopenic obesity. Sarcopenic obesity has been shown to carry the cumulative risk of low SMM and high fat mass [30]. In clinical practice, the start of nutritional support in cancer patients is mainly guided by their body weight at presentation and loss of body weight prior to treatment rather than body composition, i.e., the amount of SMM and fat mass. This approach may result in underdiagnosis of patients in need of nutritional support.

Although previous studies have shown that loss of SMM in patients who received cancer treatment [14,15,16,31] had significant prognostic impact on survival, in this study, a loss in SMM showed no prognostic impact for OS nor DFS. This difference may be explained by the heterogeneity in the definition of muscle mass changes, the timing of the follow-up imaging, and the type and stage of cancer and its type of treatment. Another explanation might be that in HNC dietician guidance is incorporated earlier into standard care practice than in non-HNC due to the high risk of malnutrition in patients with HNC.

The median time between follow-up imaging and pre-CRT imaging in our study was 6 months (range 5–9). In previous studies conducted in non-HNC cancer patients in which loss of SMM showed to have prognostic value, this interval ranged between 9–27 months [27] and 9–18 weeks [15]. In addition to the difference in time interval between this study and previous studies, there was also a difference in the investigated study population. In this study, we included patients who received cisplatin-based CRT in a curative setting. In a previous study, a prognostic impact of loss of SMM after CRT for decreased survival was also demonstrated in the palliative setting [14]. Timing of baseline and follow-up SMM assessment may influence the prognostic value of loss of SMM. Nevertheless, this study and the previous studies conducted on muscle wasting in cancer patients all underline the finding of significant SMM changes, which itself is an interesting finding which needs more standardized and prospective research to evaluate its value for treatment outcomes and prognosis in patients with cancer.

For patients with HNC, the frequent use of CT and MRI imaging for staging, evaluation, and surveillance provides the opportunity to measure SMM without additional patient burden or costs. SMM assessments can serve as an objective and clinical measure of patient nutritional status and physical vulnerability and can be used to predict treatment outcomes in patients with cancer. SMM can be objectively and reliably measured and is a potentially modifiable risk factor. An increased understanding of the underlying mechanisms of the negative prognostic effects of low SMM in patients with cancer is crucial in order to innovate and to improve current treatment strategies and eventually treatment outcomes. Commonly proposed strategies include combination of high-protein nutritional support, exercise, and pharmacological interventions. Use of an intervention program, which includes nutrition support and high-intensity exercise, is probably an ideal option for patients with low SMM [31,32].

Our study had some limitations. Firstly, due to the retrospective design of the study, not all information on potential confounding variables could be retrieved, such as life-style measures including nutritional support and physical exercise. Nutritional support during CRT may influence the observed changes in SMM and their prognostic value. Secondly, we used routinely performed baseline and follow-up imaging and the time between the baseline and follow-up imaging therefore varied between patients. This may have inherently led to bias of the results. Thirdly, muscle function and muscle strength were not measured in this study. However, these measures are also important in functional depletion and should be investigated further. Another limitation is that due to the discrepancy of males and females in this study, 71 females and 164 males respectively, we did not calculate gender-specific cutoff values for low SMM. SMM may however differ between males and females. Further studies should be performed to determine standardized gender-specific cutoff values for low SMM in patients with head and neck cancer.

In conclusion, this study is the first to evaluate longitudinal SMM changes in patients with LA-HNC treated with cisplatin-based CRT and the first to identify risk factors for loss of SMM. Loss of SMM after CRT occurs in majority of LA-HNC patients. Obese patients were at increased risk for experiencing loss of SMM.

## Figures and Tables

**Figure 1 jcm-10-01762-f001:**
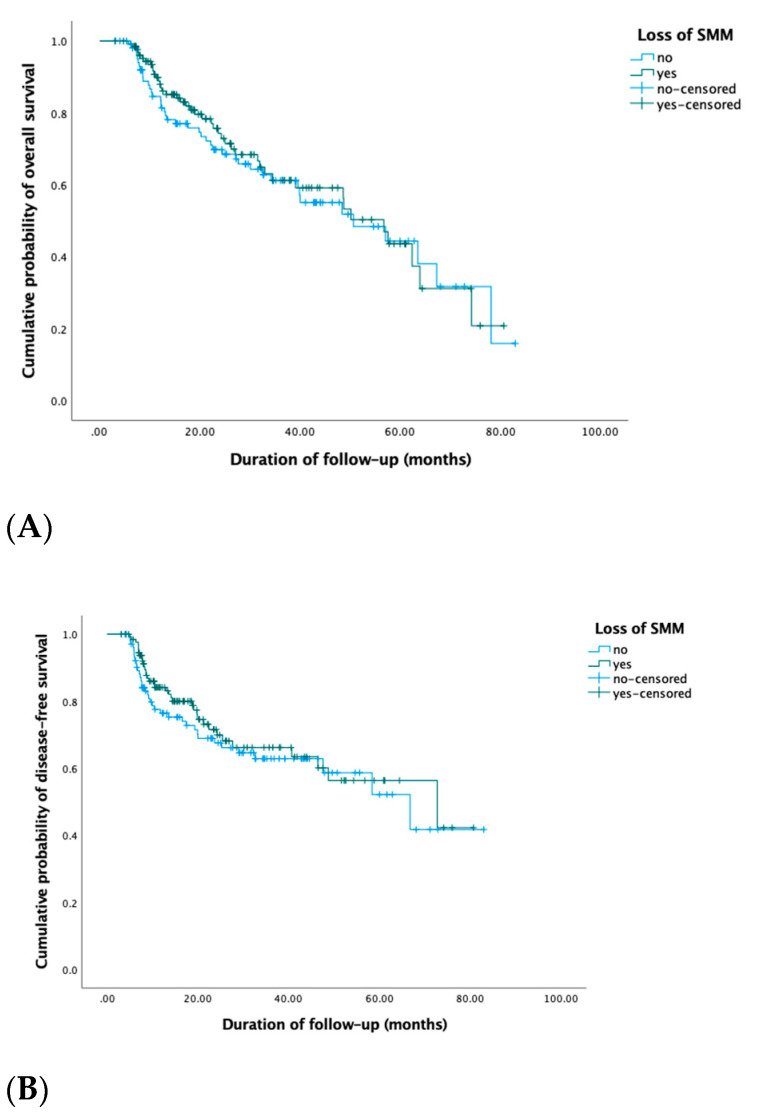
Kaplan–Meier OS (**A**) and DFS (**B**) curves of patients with stable SMM versus muscle changes (muscle loss and muscle gain) showed no significant differences in OS (log-rank Chi-square 0.1, *p* = 0.8) nor DFS (log-rank Chi-square 0.5, *p* = 0.5).

**Figure 2 jcm-10-01762-f002:**
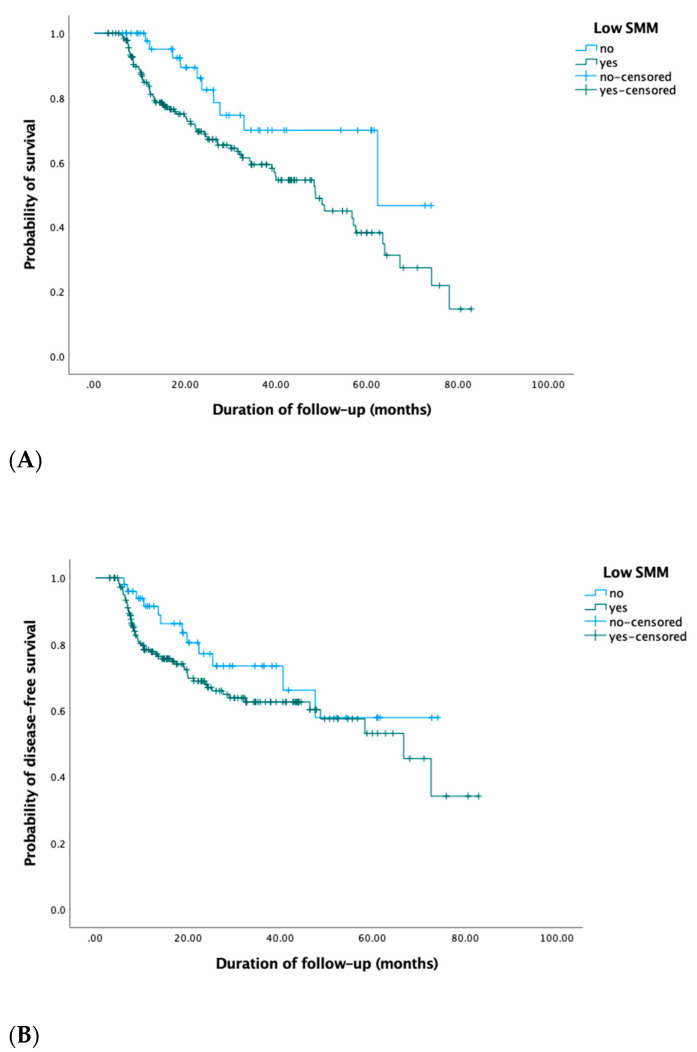
Kaplan–Meier OS (**A**) and DFS (**B**) curves of patients with low SMM at diagnosis showed decreased OS (log-rank Chi-square 5.1, *p* = 0.02) and DFS (log-rank Chi-square 2.0, *p* = 0.2) compared to patients without low SMM, although this finding was only statistically significant for OS and not DFS.

**Table 1 jcm-10-01762-t001:** Study population characteristics.

Characteristics	Total (*n* = 235)	Low SMM (*n* = 185)	Without low SMM (*n* = 50)	*p*-Value
	N (%)	N (78.7%)	N (21.3%)	
Gender Male Female	164 (69.8) 71 (30.2)	115 (62.2) 70 (37.8)	49 (98) 1 (2)	**<0.01**
Age diagnosis (years) (mean, SD)	58.6 (8.0)	59.1 (7.9)	56.8 (8.2)	0.07
BMI (kg/m^2^) <18.5 18.5–24.9 25.0–29.9 ≥30	21 (8.9) 117 (49.8) 65 (27.7) 32 (13.6)	21 (11.4) 113 (61.1) 40 (21.6) 11 (5.9)	0 (0) 4 (8) 25 (50) 21 (42)	**<0.01**
ACE-27 score None Mild Moderate Severe	50 (21.3) 115 (48.9) 53 (22.6) 17 (7.2)	42 (22.7) 87 (47) 43 (23.2) 13 (7)	8 (16) 28 (56) 10 (20) 4 (8)	0.64
Performance status ECOG 0 ECOG 1 ECOG ≥ 2 Missing	63 (26.8) 111 (47.2) 27 (11.5) 34 (14.5)	49 (26.5) 89 (48.1) 21 (11.4) 26 (14.1)	14 (28) 22 (44) 6 (12) 8 (16)	0.96
Smoker No Current/former	43 (18.3) 192 (81.7)	30 (16.2) 155 (83.8)	13 (26) 37(74)	0.15
Alcohol use No Yes Smoker and alcohol use No Yes	40 (17) 195 (83) 60 (25.5) 175 (74.5)	32 (17.3) 153 (82.7) 44 (23.8) 141 (76.2)	8 (16) 42 (84) 16 (32) 34 (68)	0.84 0.27
Albumin (g/L) (mean, SD)	39.8 (4.6)	40.0 (4.7)	39.8 (4.6)	0.8
Tumor site Oral cavity Oropharynx *HPV–* *HPV+* *HPV unknown* Nasopharynx Hypopharynx Larynx Paranasal sinus Unknown primary	83 (35.3) 73 (31.1) *44 (60.3)* *21 (28.8)* *8 (11.0)* 19 (8.1) 32 (13.6) 10 (4.3) 10 (4.3) 8 (3.4)	66 (35.7) 57 (30.8) *36 (63.2)* *14 (24.6)* *7 (12.3)* 14 (7.6) 27 (14.6) 8 (4.3) 7 (3.8) 6 (3.2)	17 (34) 16 (32) *8 (50.0)* *7 (9.6)* *1 (6.3)* 5 (10) 5 (10) 2 (4) 3 (6) 2 (4)	0.98
TNM stage III IV	40 (17.0) 195 (83.0)	31 (16.8) 154 (83.2)	9 (18.0) 41 (82)	0.84
CRT setting Primary Adjuvant	166 (70.6) 69 (29.4)	126 (68.1) 59 (31.9)	40 (80) 10 (20.0)	0.12

*p*-Values printed in bold were significant at a 0.05 level, *p*-values printed in italics were near statistical significance at a 0.05 level, SMM: Skeletal muscle mass, BMI: Body mass index, ACE-27: Adult Comorbidity Evaluation 27, ECOG: Easter Cooperative Oncology Group, HPV: Human papilloma virus, TNM: Tumor, node, metastasis, CRT: Chemoradiotherapy.

**Table 2 jcm-10-01762-t002:** Changes in SMM after CRT.

	Stable	Moderate Loss	Moderate Gain	Large Loss	Large gain
Change in SD	±1 SD	≥1 SD to <2 SD	≥1 SD to <2 SD	≥2 SD	≥2 SD
SMA range (cm^2^)	>24.33 to <42.45	≤24.33 to >18.22	≥42.45 to <51.56	≤18.22	≥51.56
*n* (%)	91 (38.7)	129 (54.9)	13 (5.5)	1 (0.4)	1 (0.4)

SMM: Skeletal muscle mass, CRT: Chemoradiotherapy, SD: Standard deviation, SMA: Skeletal muscle area.

**Table 3 jcm-10-01762-t003:** Characteristics of patients with stable SMM versus patients with loss and gain of SMM.

Characteristics	Stable *n* = 91 41.2%	Muscle Loss *n* = 130 90.3%	Muscle Gain *n* = 14 9.7%	*p*-Value
	*n* (%)	*n* (%)	*n* (%)	
Gender Male Female	63 (69.2) 28 (30.8)	89 (68.5) 41 (31.5)	12 (85.7) 2 (14.3)	0.4
Age > 60 years No Yes BMI (kg/m^2^) <18.5 18.5–24.9 25.0–29.9 ≥30 ACE-27 score None Mild Moderate Severe Performance ECOG 0 ECOG 1 ECOG ≥ 2 Unknown Smoking No Former Current Alcohol No Current/former	45 (49.5) 46 (50.5) 14 (15.4) 50 (54.9) 19 (20.9) 8 (8.8) 18 (19.8) 48 (52.7) 19 (20.9) 6 (6.6) 24 (26.4) 48 (52.7) 8 (8.8) 11 (12.1) 18 (19.8) 25 (27.5) 48 (52.7) 18 (19.8) 73 (80.2)	72 (55.4) 58 (44.6) 6 (4.6) 60 (46.2) 40 (30.8) 24 (18.5) 29 (22.3) 64 (49.2) 30 (23.1) 7 (5.4) 37 (28.5) 53 (40.8) 19 (14.6) 21 (16.2) 24 (18.5) 37 (28.5) 69 (53.1) 2 (14.3) 12 (85.7)	7 (50) 7 (50) 7 (50) 1 (7.1) 6 (42.9) 0 (0) 3 (21.4) 3 (21.4) 4 (28.6) 4 (28.6) 2 (14.3) 10 (71.4) 0 (0) 2 (14.3) 1 (7.1) 3 (21.4) 10 (71.4) 20 (15.4) 110 (84.6)	0.7 **0.01** 0.7 0.2 0.7 0.7
Tumor site Oral cavity Oropharynx Nasopharynx Hypopharynx Larynx Paranasal sinus Unknown primary TNM stage III IV	30 (33) 22 (24.2) 9 (9.9) 16 (17.6) 4 (4.4) 3 (3.3) 7 (7.7) 14 (15.4) 77 (84.6)	46 (35.4) 46 (35.4) 10 (7.7) 14 (10.8) 6 (4.6) 7 (5.4) 1 (0.8) 26 (20) 104 (80)	7 (50) 5 (35.7) 0 (0) 2 (14.3) 0 (0) 0 (0) 0 (0) 0 (0) 14 (100)	0.2 0.3

*p*-Values printed in bold were significant at a 0.05 level. SMM: Skeletal muscle mass, BMI: Body mass index, ACE-27: Adult Comorbidity Evaluation 27, ECOG: Easter Cooperative Oncology Group, HPV: Human papilloma virus, TNM: Tumor, node, metastasis.

**Table 4 jcm-10-01762-t004:** Logistic regression analysis: Predictors for SMM loss.

	Univariate Analysis		Multivariate Analysis	
	OR (95% CI)	*p*-Value	OR (95% CI)	*p*-Value
Age (years)	1.0 (1.0–1.0)	0.8		
Gender Female Male	Ref. 0.9 (0.5–1.5)	0.6		
BMI (kg/m^2^) 18.5–24.9 <18.5 25–29.9 ≥30 Smoking No Yes Alcohol use No Yes Performance status ECOG 0 ECOG 1 ECOG ≥ 2 Unknown Albumin (mmol/L)	Ref. 0.4 (0.1–1.0) 1.5 (0.8–2.8) 2.9 (1.2–6.9) Ref. 1.0 (0.5–1.9) Ref. 1.3 (0.7–2.6) Ref. 0.6 (0.3–1.2) 1.7 (0.6–4.4) 1.1 (0.5–2.7) 1.1 (1.0–1.1)	0.06 0.2 **0.02** 0.9 0.5 0.2 0.3 0.8 0.2	Ref. 0.4 (0.1–2.0) 1.3 (0.7–2.5) 3.6 (1.4–9.3)	0.1 0.4 **<0.01**
ACE-27 None Mild Moderate Severe Tumor localization Oral cavity Oropharynx Nasopharynx Hypopharynx Larynx Paranasal sinus Unknown primary HPV status Negative Positive Unknown CRT setting Primary Adjuvant Dose-limiting toxicity No Yes Cumulative chemotherapy dose <300 mg ≥300 mg Weight loss during CRT None <10% ≥10%	Ref. 1.0 (0.6–1.8) 0.9 (0.4–2.1) 0.5 (0.2–1.5) Ref. 1.4 (0.7–2.6) 0.9 (0.3–2.4) 0.6 (0.3–1.4) 1.2 (0.3–4.6) 1.9 (0.5–7.8) 0.1 (0.01–0.9) Ref. 0.8 (0.3–2.2) 1.7 (0.3–9.5) Ref. 1.2 (0.7–2.1) Ref. 1.2 (0.7–2.0) Ref. 0.9 (0.5–1.6) Ref. 1.4 (0.7–2.5) 2.8 (0.9–8.8)	0.8 0.9 0.2 0.4 0.8 0.3 0.8 0.4 **0.047** 0.6 0.5 0.6 0.5 0.6 0.3 0.08	Ref. 1.7 (0.9–1.4) 0.9 (0.3–2.6) 0.7 (0.3–1.7) 0.9 (0.2–3.6) 1.9 (0.4–8.4) 0.1 (0.1–1.0) Ref. 1.3 (0.7–2.4) 2.6 (0.8–8.5)	0.1 0.8 0.5 0.9 0.4 0.05 0.5 0.1

*p*-Values printed in bold were significant at a 0.05 level. SMM: Skeletal muscle mass, HR: Hazard ratio, SD: Standard deviation, BMI: Body mass index, ACE-27: Adult Comorbidity Evaluation 27, ECOG: Easter Cooperative Oncology Group, HPV: Human papilloma virus, TNM: Tumor, node, metastasis, CRT: Chemoradiotherapy.

**Table 5 jcm-10-01762-t005:** Proportional Cox regression analysis: Prognostic factors for overall survival.

	Univariate Analysis		Multivariate Analysis	
	HR (95% CI)	*p*-Value	HR (95% CI)	*p*-Value
Age (years)	1.0 (1.0–1.0)	0.2		
Low SMM No Yes	Ref. 2.1 (1.1–4.1)	**0.03**	Ref. 4.3 (0.6–32.6)	0.2
Gender Female Male	Ref. 1.1 (0.7–1.7)	0.7		
BMI (kg/m^2^) 18.5–24.9 <18.5 25–29.9 ≥30	Ref. 1.1 (0.5–2.2) 0.7 (0.4–1.1) 0.9 (0.5–1.7)	0.9 0.1 0.7		
ACE–27 None Mild Moderate Severe	Ref. 1.1 (0.6–1.8) 1.5 (0.8–2.8) 1.1 (0.4–2.6)	1.0 0.2 0.9		
Tumor localization Oral cavity Oropharynx Nasopharynx Hypopharynx Larynx Paranasal sinus Unknown primary	Ref. 1.1 (0.7–1.9) 0.6 (0.2–1.5) 1.2 (0.7–2.2) 0.6 (0.2–2.1) 0.2 (0.03–1.4) 1.1 (0.3–4.6)	0.7 0.3 0.5 0.5 0.1 0.9		
HPV status Negative Positive Unknown	Ref. 0.1 (0.01–0.9) 0.3 (0.04–2.2)	**0.04** 0.2	Ref. 0.1 (0.02–1.0) 0.4 (0.05–2.7)	0.06 0.3
CRT setting Primary Adjuvant	Ref. 1.3 (0.9–2.1)	0.2		

*p*-Values printed in bold were significant at a 0.05 level. SMM: Skeletal muscle mass, HR: Hazard ratio, BMI: Body mass index, ACE-27: Adult Comorbidity Evaluation 27, HPV: Human papilloma virus, TNM: Tumor, node, metastasis, CRT: Chemoradiotherapy.

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
