# Peer review of "Patterns, Predictors, and Prognostic Value of Skeletal Muscle Mass Loss in Patients with Locally Advanced Head and Neck Cancer Undergoing Cisplatin-Based Chemoradiotherapy"

_jcm, 2021, doi:10.3390/jcm10081762_

Round 1

Reviewer 1 Report

Comments to authors

At the beginning, I would like notice that the manuscript contains  interesting research matter, and I found it worth to be accepted for publication. Particularly noteworthy in the presented manuscript is the demonstration of a close relationship between nutritional support during CRT and changes in SMM. In my opinion, the manuscript under review is important due, to the fact that it indicates the need to provide nutritional treatment not only to patients already affected by CRT, but also after diagnosis and before starting therapy.

Author Response

At the beginning, I would like notice that the manuscript contains  interesting research matter, and I found it worth to be accepted for publication. Particularly noteworthy in the presented manuscript is the demonstration of a close relationship between nutritional support during CRT and changes in SMM. In my opinion, the manuscript under review is important due, to the fact that it indicates the need to provide nutritional treatment not only to patients already affected by CRT, but also after diagnosis and before starting therapy.

We would like to thank reviewer 1 for his evaluation.

Reviewer 2 Report

I had a great pleasure to review the paper entitled: “Patterns, predictors and prognostic value of skeletal muscle mass loss in patients with locally advanced head and neck cancer undergoing cisplatin-based chemoradiotherapy”

The authors retrospectively evaluated the change in skeletal muscle mass (SMM) among patients with locally advanced head and neck cancer patients after cisplatin-based chemoradiation therapy, and used this measurement obtained from imaging studies on two timepoints (prior to chemoradiotherapy and follow-up within 1 year after chemoradiotherapy completion), to identify its predictors and evaluate its usefulness in predicting the survival for locally advanced head and neck cancer patients who underwent chemoradiotherapy.

This is an interesting study; however, there are major issues including methodological problems that significantly limited this study.

Major issues:

  • Using a cutoff point (LSMI ≤ 43.2 cm2/m2) that is not well validated to categorize SMM that was adopted from a previous work is highly questionable and have greatly influenced and limited the results of the current study. This can be easily observed in the imbalance between groups. For example, only 3 (4.2% of the total) females didn’t have low SMM, compared to 91 (55.5% of the total) males. Another example is the difference in percentages of low SMM between pre-CRT and follow up-imaging (60% to 63%). If another cutoff point was used, the difference could have been more obvious. Also, the cutoff point in the original article was identified as a predictor of chemotherapy dose-limiting toxicity, which is not the outcome of interest in this study.
  • Grouping all head and neck cancer types together when studying the survival is problematic for the great variation in survival between different head and neck cancer types.
  • HPV status is a large drive of survival in head and neck cancer patients, and this must be considered in the survival analysis for the results to be considered to be strong.
  • Using stepwise elimination (Backward), is inappropriate. Investigating the variables should be based on a single hypothesis to be tested, which in stepwise model, this is violated. Some variables are important to include in the multivariable analysis, even if their P value in the univariate analysis is near significance (<0.1)
  • Using a cox proportional hazard model of the predictors for loss of SMM is suboptimal given the nature of the study. The time interval chosen for this step will highly depends on the follow-up imaging, which in turn, have many factors that influence the time including the condition of the patient, signs of cancer progression …. Etc. Therefore the presence or absence of low SMM is the outcome that needs to be investigated and not the time to the event.

Minor issues:

  • For some frequency comparison, Fisher exact test is more appropriate than Chi-square statistics as some of the comparison used failed the Chi-squared test assumptions (the frequency is at least 5 for at least 80% of the categories or the frequency is at least 1 for each expected category).
  • Table rows and not well aligned with the variables (though this could be a pdf conversion problem).
  • Few parts of the manuscript may need some improvement in finer points of grammar and punctuation.

Recommendations:

  • Analyzing and reporting SMM measurement as a continuous factor can be highly valuable when compared between different groups and different time points (Pre Vs. follow up imaging)
  • Identify a cutoff point specific for the current study and the outcome of interest. It would be useful as well to stratify the cutoff point for different sex.
  • Choosing a unique cancer type (For example oropharynx site only because of its inherently different survival) can better improve the homogeneity of the study population rather than grouping all the cancer types together.
  • Another approch that authors can use, is sub-analysis stratified by different cancer types (oral cavity, oropharynx and other types). In this case, HPV status can be used as an important factor to control for in the oropharynx sub-analysis.
  • A logistic regression model is more appropriate to identify predictors or risk factors for loss of SMM after CRT.
  • Variables selection should not rely on stepwise elimination. A suggestion would be including all variables with P value less than 0.1. Example is the weight, although the P value is 0.09, the 95% CI is suggestive of its importance, which is consistent with the clinical observations. If the authors are worried about collinearity of the model, this should be investigated and reported.
  • Use Fisher exact test for the comparison where the frequency is at least 5 for at least 80% of the categories or the frequency is at least 1 for each expected category.

Author Response

Reviewer 2

I had a great pleasure to review the paper entitled: “Patterns, predictors and prognostic value of skeletal muscle mass loss in patients with locally advanced head and neck cancer undergoing cisplatin-based chemoradiotherapy”

The authors retrospectively evaluated the change in skeletal muscle mass (SMM) among patients with locally advanced head and neck cancer patients after cisplatin-based chemoradiation therapy, and used this measurement obtained from imaging studies on two timepoints (prior to chemoradiotherapy and follow-up within 1 year after chemoradiotherapy completion), to identify its predictors and evaluate its usefulness in predicting the survival for locally advanced head and neck cancer patients who underwent chemoradiotherapy.

This is an interesting study; however, there are major issues including methodological problems that significantly limited this study.

We would like to thank reviewer 2 for his evaluation. See below for our answers on the major issues mentioned by reviewer 2.

Major issues:

Using a cutoff point (LSMI ≤ 43.2 cm2/m2) that is not well validated to categorize SMM that was adopted from a previous work is highly questionable and have greatly influenced and limited the results of the current study. This can be easily observed in the imbalance between groups. For example, only 3 (4.2% of the total) females didn’t have low SMM, compared to 91 (55.5% of the total) males. Another example is the difference in percentages of low SMM between pre-CRT and follow up-imaging (60% to 63%). If another cutoff point was used, the difference could have been more obvious. Also, the cutoff point in the original article was identified as a predictor of chemotherapy dose-limiting toxicity, which is not the outcome of interest in this study.

Universal cutoff points for low SMM are lacking. Therefore, we chose to use the LSMI cutoff point of 43.2 cm2/ m2 calculated in our previous work in a separate cohort of head and neck cancer. We agree with reviewer 2 that this is not ideal as this cutoff point was calculated for the endpoint chemotherapy dose-limiting toxicity. Therefore, we decided to calculate a new cohort-specific cutoff point for this specific cohort with the endpoint survival. We corrected all analysis for this new cutoff point. The new cutoff point is a 46.6 cm2/ m2, see line 126-128 in the article.

Grouping all head and neck cancer types together when studying the survival is problematic for the great variation in survival between different head and neck cancer types.

HPV status is a large drive of survival in head and neck cancer patients, and this must be considered in the survival analysis for the results to be considered to be strong.

We agree with the reviewer, we added a new table (table 5) with a proportional cox regression analysis to identify the prognostic factors for overall survival. We included HPV status in the multivariate analysis. See page 22.

Using stepwise elimination (Backward), is inappropriate. Investigating the variables should be based on a single hypothesis to be tested, which in stepwise model, this is violated. Some variables are important to include in the multivariable analysis, even if their P value in the univariate analysis is near significance (<0.1)

We agree with the reviewer and changed our analysis. See lines 156-166.

Using a cox proportional hazard model of the predictors for loss of SMM is suboptimal given the nature of the study. The time interval chosen for this step will highly depends on the follow-up imaging, which in turn, have many factors that influence the time including the condition of the patient, signs of cancer progression …. Etc. Therefore the presence or absence of low SMM is the outcome that needs to be investigated and not the time to the event.

We agree with the reviewer and changed our analysis. See lines 156-166 and table 4.

Minor issues:

For some frequency comparison, Fisher exact test is more appropriate than Chi-square statistics as some of the comparison used failed the Chi-squared test assumptions (the frequency is at least 5 for at least 80% of the categories or the frequency is at least 1 for each expected category).

We agree with the reviewer and changed our analysis. See lines 152-154 and table 1.

Table rows and not well aligned with the variables (though this could be a pdf conversion problem).

We aligned the table rows with the variables.

Few parts of the manuscript may need some improvement in finer points of grammar and punctuation.

We improved this.

Recommendations:

Analyzing and reporting SMM measurement as a continuous factor can be highly valuable when compared between different groups and different time points (Pre Vs. follow up imaging)

We choose to analyze and report SMM measurements as a categorical variable in line with previous studies using this method.

Identify a cutoff point specific for the current study and the outcome of interest. It would be useful as well to stratify the cutoff point for different sex.

We agree with the reviewer and changed our cutoff point to a cohort specific cutoff point for the outcome survival. Due to the difference in numbers of male and female patients in this cohort, we did not stratify the cutoff point for different sex.

Choosing a unique cancer type (For example oropharynx site only because of its inherently different survival) can better improve the homogeneity of the study population rather than grouping all the cancer types together.

We agree with the reviewer that different cancer types have different outcomes, especially HPV-related oropharynx carcinoma. Therefore, we choose to include the tumor site and HPV-status in the regression analysis.

Another approach that authors can use, is sub-analysis stratified by different cancer types (oral cavity, oropharynx and other types). In this case, HPV status can be used as an important factor to control for in the oropharynx sub-analysis.

We choose to include the tumor site and HPV-status in the regression analysis.

A logistic regression model is more appropriate to identify predictors or risk factors for loss of SMM after CRT.

We agree with the reviewer and choose to perform a logistic regression model.

Variables selection should not rely on stepwise elimination. A suggestion would be including all variables with P value less than 0.1. Example is the weight, although the P value is 0.09, the 95% CI is suggestive of its importance, which is consistent with the clinical observations. If the authors are worried about collinearity of the model, this should be investigated and reported.

Use Fisher exact test for the comparison where the frequency is at least 5 for at least 80% of the categories or the frequency is at least 1 for each expected category.

We agree with the reviewer and choose to include all variables with p values less than 0.1 and used the fisher exact test for the comparison where the frequency is at least 5 for at least 80% of the categories or the frequency is at least 1 for each expected category.

Reviewer 3 Report

This is nicely done study investigating the role of skeletal muscle mass loss in patients undergoing cisplatin-based chemoradiotherapy.  This study is relatively novel as not has been published on this topic in head and neck cancer and these authors build on prior work and provide a good starting point for future studies.  

  1. The inclusion study are broad across head and neck cancer sites.  
  2. The study includes patients who also underwent surgery and then received adjuvant radiation.  I don't feel like the inclusion of this group is adequately addressed and maybe should be excluded from the study.  Patients who undergo adjuvant CRT generally will be getting a lower dose of radiation to the primary site and likely is the reason the authors find the SMM loss in the adjuvant setting is not as severe as in the primary setting.  The authors need to better address this finding in the discussion or exclude this group of patients.    
  3. While statistically significant, is the change in mean SMM shown in table 2 clinically significant?  This issue is not addressed in the discussion and there may not be a benchmark.
  4. It would be interesting to look at the BMI pre and post treatment.  The authors raise the concept of sarcopenic obesity, and I think it would be helpful to know if the change in BMI is at a similar rate of the change in SMM.  Obese patients generally lose a good deal of weight during treatment and it would be helpful to try to get an estimation of what percentage of the change in BMI is related to the change in SMM.
  5. The authors do not report the gastrostomy tube rate in their patient population.  This is a major shortcoming and needs to be added to the paper.  Many patients get a Gtube during treatment and the reader needs to know if this has an impact on SMM.  

Author Response

Reviewer 3

This is nicely done study investigating the role of skeletal muscle mass loss in patients undergoing cisplatin-based chemoradiotherapy.  This study is relatively novel as not has been published on this topic in head and neck cancer and these authors build on prior work and provide a good starting point for future studies. 

The inclusion study are broad across head and neck cancer sites. 

The study includes patients who also underwent surgery and then received adjuvant radiation.  I don't feel like the inclusion of this group is adequately addressed and maybe should be excluded from the study.

We agree with the reviewer and therefore we choose to include the setting of CRT (primary or adjuvant) as a variable in the analysis, including table 1, 4 and 5.

Patients who undergo adjuvant CRT generally will be getting a lower dose of radiation to the primary site and likely is the reason the authors find the SMM loss in the adjuvant setting is not as severe as in the primary setting.  The authors need to better address this finding in the discussion or exclude this group of patients.

We agree with the reviewer, when adjusting our analysis based on the recommendations of reviewer 2, we did no longer see a significant predictive impact of CRT setting for SMM loss. Therefore, we did not decide to adjust the discussion.

While statistically significant, is the change in mean SMM shown in table 2 clinically significant?  This issue is not addressed in the discussion and there may not be a benchmark.

It would be interesting to look at the BMI pre and post treatment.

We agree with the reviewer and mentioned this in the discussion, see line 230-231.

The authors raise the concept of sarcopenic obesity, and I think it would be helpful to know if the change in BMI is at a similar rate of the change in SMM.  Obese patients generally lose a good deal of weight during treatment and it would be helpful to try to get an estimation of what percentage of the change in BMI is related to the change in SMM.

We agree with the reviewer but unfortunately due to the design of this study (retrospective) not all data of BMI at the time of the second imaging was available.

The authors do not report the gastrostomy tube rate in their patient population.  This is a major shortcoming and needs to be added to the paper.  Many patients get a Gtube during treatment and the reader needs to know if this has an impact on SMM.

We agree with the reviewer but unfortunately due to the design of this study (retrospective) majority of data on nutritional support during CRT was unavailable. We did already mention this in our limitation, see line 289-290.

Round 2

Reviewer 2 Report

I thank the authors for the effort they put together to address my previous comments. I only have few minor issues that needs to be addressed:

  • Line 132 needs a citation. Please cite the reference for Williams et al. paper.
  • In the paper use <18.5 and not ≤18.5 for the underweight BMI category, consistent with your grouping of normal 18.5-24.9 for normal and the WHO category as well.
  • The chi-squared test or the Fisher exact test needs to be used as appropriate (both needs to be used here, using chi-squared test when the frequency is at least 5 for at least 80% of the categories or the frequency is at least 1 for each expected category and using the Fisher exact test when this assumption is not met).
  • Describe in the limitation or the discussion how “Universal cutoff points for low SMM are lacking.”
  • Please discuss the point about sex specific cutoff point in the limitation section.

Author Response

We would like to thanks the reviewer for the comments to improve our manuscript.

Line 132 needs a citation. Please cite the reference for Williams et al. paper.

Done accordingly.

In the paper use <18.5 and not ≤18.5 for the underweight BMI category, consistent with your grouping of normal 18.5-24.9 for normal and the WHO category as well.

Changed accordingly

The chi-squared test or the Fisher exact test needs to be used as appropriate (both needs to be used here, using chi-squared test when the frequency is at least 5 for at least 80% of the categories or the frequency is at least 1 for each expected category and using the Fisher exact test when this assumption is not met).

Changed accordingly

Describe in the limitation or the discussion how “Universal cutoff points for low SMM are lacking.”

Added accordingly

Please discuss the point about sex specific cutoff point in the limitation section.

Added accordingly
